# Fluid movements enhance creative fluency: A replication of Slepian and Ambady (2012)

**Shu Imaizumi** [1]*, **Ubuka Tagami**[2], **Yi Yang**[2]

**1** Institute for Education and Human Development, Ochanomizu University, Tokyo, Japan, **2** Graduate School of Humanities and Sciences, Ochanomizu University, Tokyo, Japan

* imaizumi.shu@ocha.ac.jp

## Abstract

Bodily movements representing abstract concepts (e.g., fluidity) can affect divergent creative thinking. A recent study showed that participants who performed fluid arm movements by tracing curved line-drawings (the fluid condition) subsequently generated a larger number of more original alternative uses for newspapers than did those who traced angular line-drawings (the non-fluid condition). This suggests that fluid movements enhance fluency and originality in divergent creative thinking. To replicate these findings, we employed the same task with a larger Japanese sample. Participants in the fluid condition generated more uses for newspapers than in the non-fluid condition, regardless of confounding variables: mood, subjective difficulty of the tracing, and daily use of newspapers. In contrast to previous findings, there were no effects on originality. Our results suggest that fluidity enacted by arm movements robustly enhances creative fluency, although other factors (e.g., culture) could interfere with its effect on originality.

**Data Availability Statement:** All relevant data are within the manuscript and its Supporting Information files.

**Funding:** This work was supported by Grant-in-Aid for Early-Career Scientists (20K20144) to SI from

## Introduction

Abstract concepts are grounded in language, emotions, and sociality [1, 2] and are metaphorically associated with concrete experiences [3–5]. For example, the concepts of interpersonal warmth and importance can be associated with physical warmth and weight, respectively. Empirical studies demonstrated that people who hold a hot drink perceive a face-to-face person as warmer than do those holding a cold drink [6], and people who hold a heavier object estimate social problems to be more serious [7]. These findings suggest that abstract concepts are also grounded in our body and sensorimotor system. Supporting this notion, bodily movements to enact a metaphor that represents an association between abstract concepts can also modulate cognition (i.e., embodied cognition). For instance, arm movements in a vertical direction, which represent a metaphorical association between vertical space and emotional valence (e.g., up–good, down–bad), can bias the recollection of emotional autobiographical memory [8] and evaluation of a previous emotional stimulus [9, 10].

Studies on embodied cognition have also revealed that bodily sensation and movements can modulate divergent creative thinking [11, 12]. For example, walking better facilitates subsequent divergent thinking than sitting [13], suggesting an effect of bodily movements *per se*

the Japan Society for the Promotion of Science (https://www.jsps.go.jp/english/e-grants/). The funder had no role in study design, data collection and analysis, decision to publish, or preparation of the manuscript.

**Competing interests:** The authors have declared that no competing interests exist.

on creativity. Moreover, the shape of the trajectory of bodily movement can provide an implicit clue to insight problem solving; for instance, swinging the arm facilitates solving problems about making a swinging pendulum with a string [14]. These findings suggest that physical components of bodily movements facilitate divergent creative thinking and problem solving.

Abstract concepts represented by somatosensation and bodily movements also modulate creativity [11, 12]. To understand abstract concepts, human cognitive as well as linguistic processes employ metaphors mapping abstract concepts onto superficially dissimilar concrete concepts [3, 5]. Recent studies have suggested that somatosensory inputs can activate such conceptual metaphorical mapping and simultaneously or subsequently affect creative thinking. For example, since openness is metaphorically associated with divergence in creative thinking, open body postures lead to better divergent creative performance than closed postures [15]. Similarly, activities involving larger areas of the facial muscles lead to broader perceptual and conceptual attention, facilitating the originality of generated ideas [16]. Furthermore, physical warmth and coldness foster relational and referential creativity, respectively [17]. More importantly for the present study, bodily movements enacting conceptual metaphors can also affect creativity. For example, the metaphorical phrase "think outside the box" refers to thinking about something regardless of given rules and constraints. Bodily movements enacting this metaphor, such as thinking while sitting outside (or inside) a box and walking freely (or alongside a path), can enhance divergent and convergent creative thinking [18].

It is theorized that human intelligence consists of fluid and crystallized aspects [19, 20]. While crystallized intelligence refers to acquired knowledge, fluid intelligence includes flexibility and fluency of thought. On the other hand, the ability to fluently generate ideas is referred to as divergent thinking and is the basis of creativity [21]. Since a physical fluid moves divergently, divergent thinking is conceptually or metaphorically associated with fluidity [22]. This association might be represented in cognitive processes, and if fluidity is enacted by bodily movements depicting a fluid trajectory, they might also activate divergent creative thinking.

Slepian and Ambady [23] further hypothesized that when the motor trajectory of an upper limb is fluid rather than rigid, the individual would be more creative. In their experiments, one participant group performed fluid arm movements by tracing curved line-drawings (i.e., the fluid condition), while the other group performed non-fluid arm movements by tracing angular line-drawings (i.e., the non-fluid condition). After the tracing tasks, both groups performed one of three tasks. These were the alternative uses task [21] in which one generates as many alternative uses of newspapers as possible in one minute (Experiment 1), the cognitive flexibility task in which one rates an exemplar word's goodness-to-fit to specific categories (Experiment 2), or the remote association task in which one generates a word associated with given words (Experiment 3) [23]. Experiments 1 and 2 assessed divergent creative thinking. In both experiments, participants in the fluid condition performed better than in the non-fluid condition, suggesting that fluidity enacted by arm movements enhances creativity. Especially with large effect sizes ($r$ of 0.40 and 0.46 for unpaired $t$-tests) in the alternative uses task of Experiment 1, fluid arm movements led to a greater number of more original uses than did non-fluid arm movements. In other words, the fluency and originality in divergent creative thinking were enhanced. However, a positive mood is known to facilitate creativity [24, 25], and subjective effortfulness in motor tasks can affect problem solving [26, 27]. Thus, these are considered as variables confounding the effect of fluidity. Slepian and Ambady [23] confirmed that self-reported mood (Experiment 1) and the subjective difficulty of tracing line-drawings (Pilot Experiment) did not differ between fluid and non-fluid arm movements. However, it remains unclear whether mood and subjective difficulty during the tracing task affect the fluency and originality, and whether they interact with the effects of fluidity.

The present study aimed to replicate the main findings of Experiment 1 of Slepian and Ambady [23]. There are four reasons to replicate their findings. First, replication studies increase certainty or promote innovation in psychological science regardless of replication success or failure [28]. It has also been argued that findings on embodied creativity require evidence for reproducibility [11]. Second, since the sample size in Slepian and Ambady [23] was not sufficiently large ($n$ = 15 for each condition), the false-positive risk in their findings cannot be ruled out [29, 30]. Third, as mentioned above, whether mood and subjective difficulty affect divergent creative thinking and interact with the effects of fluidity should be examined. Finally, we assume that daily experiences with printed newspapers may facilitate the generation of alternative uses for newspapers in relation to actual activities (e.g., housework, handicrafts) or hinder it because of functional fixedness [31, 32]. If so, experiences with printed newspapers such as subscription status could be a variable confounding the effect of fluidity.

This study employed a sufficiently large sample and conducted an experiment according to procedure in the original study [23]. Participants performed fluid or non-fluid arm movements by tracing curved or angular line-drawings, and completed the alternative uses task, mood and difficulty ratings, and demographic questions including those on newspaper subscription. We not only compared fluid and non-fluid conditions as in the original study, but also tested any (interactive) effects of confounding variables on fluency and originality in divergent creative thinking.

## Materials and methods

### Participants

The effect sizes of fluidity (unpaired $t$-test) on fluency and originality in Experiment 1 of Slepian and Ambady [23] were $d$s of 1.04 and 0.87, respectively (note $r$ converted to Cohen's $d$ [33]). The present study assumed the smaller $d$ of 0.87 in the two-sided unpaired $t$-test, an alpha of 0.05, and statistical power of 0.90. Based on *a priori* power analysis using G*Power 3.1.9.6 for Mac [34], at least 29 participants were required for each group.

In total, 63 Japanese female students participated in this study. Three participants who held a pen with their left hand were excluded from the analysis to control laterality, which may affect arm movements and their influence on divergent creative thinking (see S1 File for the analysis including all participants, which had comparable results). Finally, 30 participants were randomly assigned to the fluid condition (mean age of 17.2 years, standard deviation (SD) = 0.9) and 30 to the non-fluid condition (mean age of 17.3 years, SD = 0.7; no group difference in age, $t(58)$ = -0.48, $p$ = 0.630, $d$ = -0.13, 95% confidence interval (CI) [-0.51, 0.31]). Written informed consent was obtained from each participant. The ethics committee of Ochanomizu University approved this study.

### Materials

Three line-drawings depicting geometric and traversable shapes were printed with black ink on a white portrait A4 sheet (for examples, see Fig 1 in [23]). In the fluid condition, the line drawings consisted of curves. In the non-fluid condition, the line drawings had the same shape as the fluid condition except that they consisted of straight lines and vertices. The line drawings subtended vertically 46–54 mm and horizontally 81–86 mm in print. To control motor trajectories, the word "Start" alongside each drawing indicated where to start tracing. The other questions (see below) were printed on the other sheets and presented together with the line-drawing sheet as a single booklet.

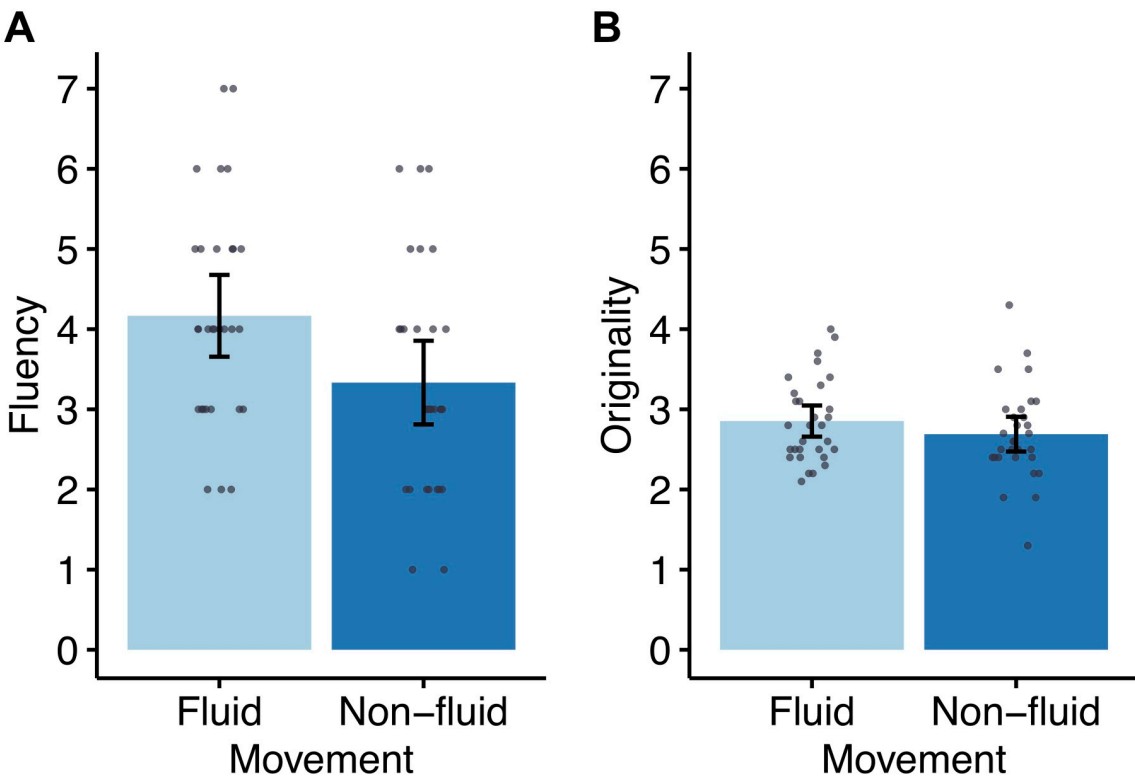

**Fig 1. Results of the alternative uses task.** (A) Fluency and (B) originality scores for the fluid and non-fluid conditions. Dots represent individual data points. Error bars denote 95% confidence interval.

### Procedures

Participants were told that the purpose of this study was to examine the relationship between hand-eye coordination and cognition, and instructed to trace the three line-drawings using a black pen at their preferred pace. Following the tracing, participants wrote down as many alternative uses for newspapers as possible within one minute (i.e., the alternative uses task [21, 23]). Subsequently, participants responded to two questions. The first asked them to rate their current mood (How do you feel right now?) on a nine-point scale ranging from 1 (very bad) to 9 (very good). The second asked them to rate the difficulty of the line tracing (How difficult was it to trace the line drawings?) on a nine-point scale ranging from 1 (not at all) to 9 (very difficult). Finally, participants reported which hand held the pen for the tracing task, their gender and age, and if they currently have a daily subscription with a printed newspaper (yes or no). After the experiment, participants were debriefed about the true purpose of this study.

### Data analysis

**Scoring creativity.** The uses reported by each participant were coded as fluency and originality according to the original study [23]. Each participant's fluency score was defined as the number of alternative uses for newspapers reported within one minute. Incomplete sentences, perhaps due to a time-out, were included as valid responses, because the authors consistently understood their meaning (one response from three participants).

We excluded repetitions from responses and identified 83 unique uses. The originality of each unique use was rated on a seven-point scale ranging from 1 (not original at all) to 7

(highly original) by 14 new Japanese volunteers (5 females; mean age of 36.5 years, SD = 10.2). Four were students in psychology, and the others were recruited from Lancers (https://www. lancers.jp/), a crowdsourcing market in Japan. The survey was conducted online using Google Forms, in which the uses were presented as rows of a table in a randomized order. Raters' scores for each use were averaged to provide the originality of the use (e.g., the lowest original- ity of 1.21 given for "Know the weather forecast"; the highest 6.21 for "Make a house"). The ratings were satisfactorily consistent among raters (McDonald's $\omega$ = 0.916). Each participant's originality was defined as the mean originality score for the uses she reported.

**Statistical analysis.** To test the effect of the fluidity of arm movements, two-sided unpaired *t*-tests comparing the fluid and non-fluid conditions were performed for the fluency, originality, mood, and subjective difficulty. When the assumption of homogeneous variance was violated based on Levene's test, Welch's *t*-test was employed. To check differences regard- ing newspaper subscription between groups, a chi-square test of the number of participants was performed. To check the interactive effects of the fluidity of arm movements and the other variables on the fluency and originality, we fitted a generalized linear model with a log link function and a Gaussian distribution. The exponential family distribution was determined based on the minimization of the Akaike information criterion. The predicted variables were the fluency and originality scores. The predictor variables were the condition (fluid or non- fluid), mood score, difficulty score, and newspaper subscription (yes or no) in addition to interactions of the condition with mood, difficulty, and subscription. The level of statistical sig- nificance was set at 0.05. Statistical analyses were performed using Jamovi 1.2.5 for Mac [35]. The dataset is available in S1 Dataset.

## Results

The fluency score was significantly higher for the fluid condition than non-fluid condition ($t(58)$ = 2.34, $p$ = 0.023, $d$ = 0.60, 95% CI [0.12, 1.55]) (Fig 1A), while the originality score did not differ between conditions ($t(58)$ = 1.16, $p$ = 0.250, $d$ = 0.30, 95% CI [-0.12, 0.45]) (Fig 1B). As summarized in Table 1, the mood and difficulty ratings and number of newspaper sub- scribers were comparable between conditions (mood: $t(49.7)$ = 0.87, $p$ = 0.386, $d$ = 0.23, 95% CI [-0.48, 1.21]; difficulty: $t(58)$ = -0.06, $p$ = 0.954, $d$ = -0.02, 95% CI [-1.18, 1.11]; subscription: $\chi^2(1)$ = 0.09, $p$ = 0.766, $V$ = 0.04).

The generalized linear models (Table 2) demonstrated that the condition (i.e., fluid or non- fluid movements) predicted fluency, while the other variables and their interactions with con- dition did not predict fluency ($R^2$ = 0.133). We did not find an effect of the condition or (inter- active) effects of the other variables on originality ($R^2$ = 0.141).

**Table 1. Descriptive statistics in the fluid and non-fluid conditions.**

|  | Fluid condition, $n$ = 30 | Non-fluid condition, $n$ = 30 |
|---|---|---|
| Fluency | 4.17 (1.37) | 3.33 (1.40) |
| Originality | 2.86 (0.52) | 2.69 (0.58) |
| Mood rating | 5.93 (1.93) | 5.57 (1.25) |
| Difficulty rating | 4.10 (2.07) | 4.13 (2.35) |
| Newspaper subscription ($n$) | 22 | 23 |

Values indicate mean (SD in parentheses) except for the number of subscribers.

**Table 2. Generalized linear models of the fluency and originality scores.**

| | Fluency | | | Originality | | |
|---|---|---|---|---|---|---|
| | **Estimate** | **z** | **p** | **Estimate** | **z** | **p** |
| Intercept | 1.32 [1.19, 1.43] | 22.12 | < 0.001 | 1.02 [0.96, 1.08] | 33.08 | < 0.001 |
| Condition | -0.29 [-0.55, -0.06] | -2.43 | 0.019 | -0.11 [-0.23, 0.01] | -1.71 | 0.093 |
| Mood rating | 0.00 [-0.08, 0.09] | 0.02 | 0.984 | 0.01 [-0.03, 0.04] | 0.35 | 0.727 |
| Difficulty rating | 0.01 [-0.03, 0.06] | 0.52 | 0.606 | -0.01 [-0.03, 0.01] | -0.79 | 0.435 |
| Subscription | -0.03 [-0.26, 0.23] | -0.26 | 0.799 | 0.00 [-0.12, 0.13] | 0.01 | 0.991 |
| Condition*Mood | 0.03 [-0.14, 0.20] | 0.39 | 0.702 | 0.02 [-0.06, 0.09] | 0.45 | 0.656 |
| Condition*Difficulty | 0.04 [-0.06, 0.13] | 0.75 | 0.457 | -0.04 [-0.08, 0.01] | -1.46 | 0.151 |
| Condition*Subscription | 0.28 [-0.20, 0.80] | 1.15 | 0.257 | 0.18 [-0.07, 0.44] | 1.41 | 0.165 |

Asterisks indicate interaction; 95% confidence interval in square brackets.

**Table 3. Results from Experiment 1 of Slepian and Ambady [23].**

| | Fluid condition, n = 15 | | Non-fluid condition, n = 15 | |
|---|---|---|---|---|
| | **Mean (SD)** | $d_{study}$ | **Mean (SD)** | $d_{study}$ |
| Fluency | 7.24 (1.21) | 2.38 | 5.65 (1.07) | 1.86 |
| Originality | 3.33 (0.42) | 0.99 | 2.85 (0.40) | 0.32 |

Values are approximate as they were drawn from the figure by a digitizer. $d_{study}$ indicates effect size (Cohen's $d$) of difference between the original and present studies.

## Follow-up analysis

The original study [23] reported larger effect sizes of fluidity on fluency and originality ($d$s = 1.04, 0.87, respectively) than did the present study ($d$s = 0.60, 0.30, respectively). To explore possible reasons for these discrepancies, we compared fluency and originality scores in both conditions between the original and present studies. Table 3 summarizes the results from Experiment 1 of the original study [23]. Since the results were only visually reported, we extracted the values from the figure using a digitizer (WebPlotDigitizer [36]). The effect sizes of the differences between the original and present studies ($d_{study}$) were calculated [37]. Substantially large values of $d_{study}$ suggest that participants in the original study performed better than did those in the present study, except for originality in the non-fluid condition.

## Discussion

This study aimed to replicate Experiment 1 of Slepian and Ambady [23], which suggested that fluid arm movements enhance fluency and originality in divergent creative thinking assessed by the alternative uses task. Our results showed that participants who traced curved line-drawings (i.e., fluid condition) subsequently generated a larger number of alternative uses of printed newspapers than did those who traced angular line-drawings (i.e., non-fluid condition). Self-reported mood after the alternative uses task, subjective difficulty of the tracing task, and daily experiences with newspapers did not predict the fluency or interact with the effect of arm movement fluidity. With a sufficiently larger sample, our results replicated and further demonstrated the facilitative effect of fluid arm movements on fluency. However, in contrast to the original study, we did not find any effects on originality.

Our results showed a selective effect of fluid arm movements on creative fluency, suggesting a link between fluency in manual movements and creative generation in metaphorical and cognitive processes. There are several domains of divergent creative thinking, such as fluency, originality, and flexibility [38]. We speculate that it is possible that a certain abstract concept selectively relates to and activates a particular aspect of divergent thinking. Indeed, another selective effect of arm movements was suggested. In a recent study [39], participants who performed arm movements tracing out large circles generated more original ideas (i.e., originality) than those who traced out small ones, but the total number of generated ideas (i.e., fluency) was comparable between conditions, suggesting that the largeness or divergence represented by arm movements could selectively influence originality. Therefore, kinematic properties such as curvature and size may determine the aspect of divergent creative thinking that is affected. We cannot rule out the possibility that the size of the drawings in our task was insufficient to affect originality.

A recent study has suggested that the effect of fluid movements on divergent creative thinking may be generalized to whole-body movements (i.e., walking) [40]. Participants who walked freely rather than along a rectangular path showed greater fluency and originality scores. The comparison between free and rectangular walks may also serve as a comparison between fluid and non-fluid movements. If so, the null effect of fluidity on originality that we found could be attributable to the difference between motor effectors (i.e., manual drawing versus whole-body walking). To speculate, fluidity could be represented more effectively by richer sensorimotor information conveyed through whole-body movements, consequently affecting divergent thinking more strongly. A direct comparison between motor effectors is needed in future studies. Furthermore, from the viewpoint of social and applied contexts, interaction with and observation of others' movements could also provide sensorimotor information that facilitates creativity, because the sensorimotor system in the brain is able to integrate observed and executed bodily movements, facilitating motor planning and execution [41]. We may speculate that joint action (e.g., line-drawing synchronously performed by individuals face to face) could enhance the effect of metaphorical movements on creativity and open a new avenue for research on embodied creativity.

The size of the significant effect of fluidity on fluency was smaller in this study than in the original study [23]. Moreover, the original study found a significant effect of fluidity on originality, while the present study did not. To explore the factors underlying these discrepancies, our follow-up analysis compared fluency and originality scores between the original and present studies and showed large values of $d_{\text{study}}$ (except for originality in the non-fluid condition), suggesting that participants in the original study showed higher fluency and originality scores. Given that the original study employed American undergraduates, cultural differences could have affected divergent creative thinking. Indeed, previous work has suggested the superior divergent thinking of an American college sample compared to a Japanese one [42]. However, note that findings on Eastern–Western differences in creativity are currently mixed [43] and even reversed in some studies [44]. We speculate that the fluidity enacted by arm movements can work more strongly for individuals with a larger capacity for divergent thinking. Indeed, a recent study suggested individual differences in embodied creativity from a different viewpoint [45], namely that the effect of arm movements (i.e., flexion or extension) on divergent thinking can be reversed depending on participants' emotional states. Thus, future studies on embodied creativity should be aware of individuals' baseline capacity for divergent thinking.

The difference in originality between the original and present studies could also be attributed to raters in different cultures. As well as participants, raters who evaluated the originality of participant's suggested alternative uses are potentially subject to the Eastern–Western difference in creativity, as that considered original varies between cultures. For example, Western

people conceptualize creativity as including intelligence, humor, and aesthetics [43, 46], while Eastern people (e.g., Chinese) also include morality [47]. Thus, different criteria applied by American and Japanese raters could influence the originality scores and lead to the different effects of fluid arm movements found in the original and present studies.

This study included only female participants, while 63% of the participants in the original study were female [23]. It could be argued that a gender effect contaminated the results. The current literature on gender difference in creativity is mixed [48]; for example, while one study reported that fluency and originality assessed through the alternative uses task did not differ between gender [49], others suggest gender differences [50]. Thus, it is difficult to attribute the different findings to different gender ratios.

The present study has two noteworthy limitations. First, like Slepian and Ambady [23], we did not include a baseline condition (e.g., without arm movements). Thus, it remains unclear whether fluid arm movements increase or non-fluid arm movements decrease fluency in divergent thinking. Second, we did not examine flexibility in divergent thinking, which was investigated in the original study. Future studies should also replicate the effect of fluid movements on flexibility.

To conclude, fluid arm movements enhanced fluency in divergent creative thinking, even when controlling confounding variables (e.g., mood). However, we did not find an effect of fluid movements on originality. Thus, we partially replicated Experiment 1 of Slepian and Ambady [23]. Given the differences in the sample of both studies (e.g., ethnicity and gender), the effect on fluency may be robust and generalizable.

## Supporting information

**S1 File. Supplementary results including all participants.**
(PDF)

**S1 Dataset. Raw data from the experiment.**
(XLSX)

## Acknowledgments

The authors thank Claudia Repetto and two anonymous reviewers for their comments on an early version of this manuscript.

## Author Contributions

**Conceptualization:** Shu Imaizumi.

**Formal analysis:** Shu Imaizumi.

**Investigation:** Shu Imaizumi, Ubuka Tagami, Yi Yang.

**Methodology:** Shu Imaizumi.

**Writing – original draft:** Shu Imaizumi.

**Writing – review & editing:** Ubuka Tagami, Yi Yang.

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
