## [Decision Letter · Decision Letter 0]

25 May 2020

PONE-D-20-05956

Fluid movements enhance creative fluency: A replication of Slepian and Ambady (2012)

PLOS ONE

Dear Dr. Imaizumi,

Thank you for submitting your manuscript to PLOS ONE. After careful consideration, we feel that it has merit but does not fully meet PLOS ONE’s publication criteria as it currently stands. Therefore, we invite you to submit a revised version of the manuscript that addresses the points raised during the review process.

We would appreciate receiving your revised manuscript by Jun 13 2020 11:59PM. To enhance the reproducibility of your results, we recommend that if applicable you deposit your laboratory protocols in protocols.io, where a protocol can be assigned its own identifier (DOI) such that it can be cited independently in the future. For instructions see: http://journals.plos.org/plosone/s/submission-guidelines#loc-laboratory-protocols

We look forward to receiving your revised manuscript.

Kind regards,

Alessandro Antonietti

Academic Editor

PLOS ONE

Additional Editor Comments (if provided):

As you can see, the reviewers' comments are mixed.

By trying to do a balance, my feeling is that only a minor revision is needed since the reviewer who suggested "major revision" indeed failed to stress heavy limits in the manuscript and just suggested improvements which can be rather easily addressed. Since literature about embodied creative cognition is not so broad, I suggest to take into account and mention all the few studies which have been conducted on such a topic. So, I suggest these further papers, beside those reported by Reviewer 3:

Andolfi, V. R., Di Nuzzo, C., & Antonietti, A. (2017). Opening the mind through the body: The effects of posture on creative processes. Thinking Skills and Creativity, 24, 20-28. doi: 10.1016/j.tsc.2017.02.01

Friedman, R. S., Fishbach, A., Förster, J., & Werth, L. (2003). Attentional priming effects on creativity. Creativity Research Journal, 15, 277–286. doi:10.1080/10400419.2003.9651420

Ijzerman, H., Leung, A. K.-Y., & Ong, L. (2014). Perceptual symbols of creativity: Coldness elicits referential, warmth elicits relational creativity. Acta Psychologica, 148, 136–147. doi:10.2139/ssrn.2235561

(I exclude papers about the effects of physical activity on creativity, which is an issue not so central to what the authors addressed in the paper).

Reviewers' comments:

Reviewer's Responses to Questions

**Comments to the Author**

1. Is the manuscript technically sound, and do the data support the conclusions?

Reviewer #1: Yes

Reviewer #2: Yes

Reviewer #3: Yes

2. Has the statistical analysis been performed appropriately and rigorously? 

Reviewer #1: Yes

Reviewer #2: Yes

Reviewer #3: Yes

3. Have the authors made all data underlying the findings in their manuscript fully available?

Reviewer #1: Yes

Reviewer #2: Yes

Reviewer #3: Yes

4. Is the manuscript presented in an intelligible fashion and written in standard English?

Reviewer #1: Yes

Reviewer #2: Yes

Reviewer #3: Yes

5. Review Comments to the Author

Reviewer #1: The present replication study presents a faithful replication of a prior study. The methods are described in sufficient detail and the analyses appear performed appropriately and rigorously. The discussion of the results is thorough. I appreciated the several points of discussion and the careful review and integration of relevant literature.

Everything looks in great shape. For a replication study, all the needed elements are here.

The only change I would suggest is how the authors discuss Thoma & Lleras, 2009 (reference 11). Here, the authors write, "For example, insight problem solving and divergent thinking are facilitated by swinging compared to stretching arm movements [11]" but this does not capture the reason why. In the specific study the authors cite here, participants were tasked with completing a specific insight puzzle: Maier's two-string problem. To solve the puzzle, the participant has to recognize that the string can be used to make a *swinging* pendulum. That's the key insight to solving the puzzle, and when participants practice this movement, they were able to figure it out more quickly. In this study, the shape of the movement helped because it physically matched an element of the solution.

Reviewer #2: The manuscript reports on a study replicating the paradigm proposed by Slepian and Ambady (2012), in which it was demonstrated that performing fluid movements improve creative thinking and originality.

In the present study previous findings were partially confirmed: participants performing curve movements (fluid condition) were able to generate a larger number of alternative uses for newspapers than those in the non-fluid condition.

The paper is concise and well written, therefore it is easy to follow. However I have few suggestions to improve it:

- sometimes concision hinders clarity, for instance I would recommend to explain more the studies reported in lines 41-43

-lines 85-86: it is not clear to me why authors state that it is still unsure, based on the original study, whether mood and task difficulty impacted fluidity and originality. It should be better specified in what respect the original paper missed the point (as in it previously said that Slepian and Ambady already considered these possible confounding variables)

- Figure 1 is very “old style” and inconsistent with the strong statistical analyses conducted. It could be replaced by a jittered plot that is more informative about the sample distribution (Jamovi creates nice ones!)

- I would move the lines 212-217 and Table 3 to the results section. It is not usual to see such results details in the discussion. In the discussion it is worth commenting on the meaning of the differences found between cultures.

Reviewer #3: Review:

This manuscript is original and quite interesting. I applaud the attempt to replicate a previous study.

As you know replication crisis is one of the most important issue in the psychological literature, so It is very welcome a study which wants to strength some results and to control for confounding variables, especially by taking into account cross-cultural differences. Objective of the manuscript is the relation between the bodily fluency (fluent movement vs non fluent movements) and the conceptual originality and fluency.

However I have some comments:

Abstract:

• sentence line 10, I recommend to the authors to be less general and to make more explicit the statement.

Introduction

• Lines 24-25 : I would consider also the fact the multiple representation views of Abstract Concept. Abstract concepts involve also linguistic, emotional and social experiences as well as internal experiences. See: “Borghi, A. M., Barca, L., Binkofski, F., & Tummolini, L. (2018). Varieties of abstract concepts: development, use and representation in the brain. Philosophical Transactions of the Royal Society B: Biological Sciences, 373(1752).” and

Borghi, A. M., & Binkofski, F. (2014). Words as social tools: An embodied view on abstract concepts (Vol. 2). Springer New York.

• I would not neglect the social component of abstract concepts as well, compared to the concrete one. See Fini, C., & Borghi, A. M. (2019). Sociality to Reach Objects and to Catch Meaning. Frontiers in psychology, 10.

• Lines 29-30: Please specify that you are talking about abstract concepts

• Could you please better clarify and make explicit the theoretical shift from the fact the bodily movements affect insight problem solving and divergent thinking to the fact that concepts represented by bodily movements can affect creativity. The passage has to be smooth for the reader.

• Lines 42-43 I do not agree with the fact that arm flexion movements represents an approaching behavior: see De Houwer, Crombez, Baeyens, & Hermans, 2001, Krieglmeyer, De Houwer, & Deutsch, 2011, Markman & Brendl, 2005, Seibt, Neumann, Nussinson, & Strack, 2008; Schneirla, 1959 according to which positive/negative stimuli lead to approach/avoidance tendencies understood as tendencies to decrease/increase distance irrespective of the specific muscle movements involved.

• Line 49-50 Too nebulous these sentences

• Line 54 Please specify what you mean for “fluid movements” already

• Line 61 “The generation of many alternative uses of newspapers” is a neuropsychological task which can be considered under which label?

Statistical analysis

• I am not familiar with the method implemented by the authors to analyze to predict the fluency and the originality scores. Can you better explain it? Clarify that you have a fluency dependent variable and a Fluency predictor variable, it is not clear.

• Why did you not implement e backward stepwise procedure to compare different models and choose the best in the goodness of fit? I mean why did you not perform a Likelihood

ratio test to compare different models varied for the complexity of the random and the fixed

effects structures?

Results

• Can you report exactly the means of the Fluency score?

Discussion

• Please speculate more on the link about the bodily fluency and the conceptual fluency.

Could you please enrich the discussion with relevant embodied literature?

I also recommend to enrich the embodied literature in these papers:

Era, V., Boukarras, S., & Candidi, M. (2019). Neural correlates of action monitoring and mutual adaptation during interpersonal motor coordination: Comment on" The body talks: Sensorimotor communication and its brain and kinematic signatures" by G. Pezzulo et al. Physics of life reviews, 28, 43.

Kuo, C. Y., & Yeh, Y. Y. (2016). Sensorimotor-conceptual integration in free walking enhances divergent thinking for young and older adults. Frontiers in psychology, 7, 1580..

6. PLOS authors have the option to publish the peer review history of their article (what does this mean?). If published, this will include your full peer review and any attached files.

Reviewer #1: No

Reviewer #2: Yes: Claudia Repetto

Reviewer #3: No

---

## [Author Response · Author response to Decision Letter 0]

12 Jun 2020

For our responses, please see the attached.

---

## [Editor Report · Decision Letter 1]

15 Jul 2020

Fluid movements enhance creative fluency: A replication of Slepian and Ambady (2012)

PONE-D-20-05956R1

Dear Dr. Imaizumi,

We’re pleased to inform you that your manuscript has been judged scientifically suitable for publication and will be formally accepted for publication once it meets all outstanding technical requirements.

Kind regards,

Alessandro Antonietti

Academic Editor

PLOS ONE
---

## [Editor Report · Acceptance letter]

16 Jul 2020

PONE-D-20-05956R1 

Fluid movements enhance creative fluency: A replication of Slepian and Ambady (2012) 

Dear Dr. Imaizumi:

I'm pleased to inform you that your manuscript has been deemed suitable for publication in PLOS ONE. Congratulations! Your manuscript is now with our production department. 

Kind regards, 

on behalf of

Prof. Alessandro Antonietti 

Academic Editor

PLOS ONE